

# Are mean vertical velocities from PMSE a good representation of mean vertical winds?

Nikoloz Gudadze [1,2], Gunter Stober [1], and Jorge L. Chau [1]

[1]Leibniz Institute of Atmospheric Physics at the University of Rostock, Kühlungsborn, Germany
[2]Abastumani Astrophysical Observatory at Ilia State University, Abastumani, Georgia

**Correspondence:** Nikoloz Gudadze (gudadze@iap-kborn.de)

**Abstract.** Mean vertical velocity measurements obtained from Radars at polar latitudes using Polar Mesosphere Summer Echoes (PMSE) as an inert tracer have been considered as non-representative of the mean vertical winds over the last couple of decades. PMSEs observed with the Middle Atmosphere Alomar Radar System (MAARSY) over Andøya, Norway (69.30°N, 16.04°E) during summers of 2016 and 2017 are used to derive mean vertical winds in the upper mesosphere. The 3D vector wind components (zonal, meridional and vertical) are based on a Doppler beam swinging experiment using 5-beam directions (one vertical and four obliques). The 3D wind components are computed using a recently developed wind retrieval technique. The method includes full non-linear error-propagation, spatial and temporal regularization as well as beam pointing corrections and angular pointing uncertainties. Measurement uncertainties are used as weights to obtain seasonal weighted averages and characterize seasonal mean vertical velocity. Weighted average values of vertical velocities reveal a weak upward behaviour at altitudes 84-87 km after eliminating the influence of ice falling speed. At the same time, a sharp decrease/increase in the mean vertical velocities at the lower/upper edges of the summer mean altitude profile prevails, which are attributed to the sampling issues of PMSE due to disappearing of the target corresponding to the certain regions of motions and temperatures. Thus the mean vertical velocities can be biased with decrease up-/down-ward velocity measurements at lower/upper edges, while at the main central region the obtained mean vertical velocities are consistent with expected values of mean vertical winds after considering ice particle sedimentation.

*Copyright statement.* TEXT

# 1 Introduction

Knowledge of the neutral wind behaviour (or motion of the air) is one of the main interest in atmospheric sciences from troposphere up to thermosphere to investigate various dynamical processes. Improving temporal and spatial resolution of wind vector components is an important challenge for different observational techniques and data analysis. Such improvement are of particular importance in the vertical component, since expected mean vertical velocities are in the range of a few centimeters per second, thus require more sophisticated observations as well as data analysis to obtain a reliable mean climatology.



Mean vertical winds are known to be an important contributor to the thermal structure of the middle atmosphere and is related to the dynamical processes of the global seasonal pole-to-pole circulation (Garcia and Solomon, 1985; Becker, 2012; Smith, 2012). Global Circulation Models show a mean summer upwelling from the lower mesosphere up to the mesopause heights at extra-tropical latitudes (May-August and November-February at Northern and Southern hemispheres respectively). The impact

of the mean vertical motion on the mesospheric temperatures is due to the expansion of the uprising air masses causing adiabatic cooling of the upper mesospheric altitudes in summer and vice versa during the winter seasons. The underlying dynamical processes where firstly proposed by Lindzen (1981) and later parametrized by Holton (1982).

The background mechanism is the influence on the mean zonal flow caused by breaking of gravity waves (GWs) in the upper mesosphere. GWs are usually generated in the lower atmosphere, and propagate upward carrying momentum and energy.

Mainly due to the solar influence and chemical processes, westward zonal winds are dominant in the mesosphere during the summer season (Becker, 2012) below 90 km. As a rule, propagating GWs are filtered out or reflected by co-directed background winds. Thus mainly eastward propagating GWs, generated in the lower atmosphere, can reach the upper mesospheric altitudes. Deposition of momentum end energy after their breaking creates wave forcing as an eastward drag. Such forcing decelerates the westward zonal wind on the corresponding heights and causes widely observed wind reversal at the lowest thermospheric

altitudes (Yuan et al., 2008; Jacobi, 2011; Wilhelm et al., 2017). The Coriolis effect from the equatorward meridional winds compensates the strong zonal wave forcing (Holton and Alexander, 2013). Consequently, upwelling of air from lower altitudes is required to satisfy the mass continuity considering the Boussinesq approximation, which holds for an incompressible atmosphere.

Propagating GWs and various turbulent process are known as a source of the common vertical motion field in the mesopause

region. One should note that the expected magnitude of the mean upwelling is in the order of a few centimetres per second, thus, very difficult to be observed. Proper statistical analysis and error estimation are needed to ensure significant wind estimates with such an accuracy. In particular, as the short time fluctuations (less than 10 minutes) of the vertical motion usually reach several meters per second.

The main tracer to obtain vertical velocities in the mesosphere during the summer seasons at polar latitudes are Polar Meso-

spheric Summer Echoes (PMSEs) measured by radars (Balsley and Riddle, 1984; Hoppe and Fritts, 1995a; Czechowsky and Rüster, 1997). PMSEs are the very strong echoes from the turbulent atmosphere at 80-90 km altitude region. Charged ice particles are responsible for slowing down the electron diffusivity and enhance the high reflectivity usually observed at VHF-frequencies, as a result of changes in the refractive index mainly at the Bragg scale of the Radar wavelength (Rapp and Lübken, 2004; Rapp et al., 2008; Kirkwood et al., 2008). PMSEs are admitted as an inert tracer and used to study the ambient dynamical

processes in the summer polar mesosphere (Stober et al., 2013, 2018b). Detailed progress in PMSE physics since their first observations by Czechowsky et al. (1979) and later Ecklund and Balsley (1981) one can find in the following review papers (Cho and Röttger, 1997; Rapp and Lübken, 2004).

Required ice clouds with particle sizes from a few to sometimes 100 nm form under dramatic low temperatures at polar mesospheric altitudes (Lübken, 1999). They are known as PMC (polar mesospheric clouds) or NLC (Noctilucent clouds). NLC are

observed from the middle latitudes up to the polar regions (Fiedler et al., 2011; Baumgarten and Fritts, 2014; Fritts et al., 2014;



Hervig et al., 2016; Gerding et al., 2018). Kiliani et al. (2013) have shown that the westward zonal transport of the NLC is connected to cloud forming under GW influence. The horizontal advection of the PMSE patches has also been recently observed using bi-static radar measurements by Chau et al. (2018).

Besides theoretical and model expectations (Lindzen, 1981; Holton, 1982; Garcia and Solomon, 1985), the mean values of the
vertical velocities measured by radars have shown contradicting, downward behaviour (Balsley and Riddle, 1984; Fritts et al., 1990; Hoppe and Fritts, 1995a).

The first study of monthly mean values of vertical velocities from vertically measured radial (Doppler) velocities were presented in Balsley and Riddle (1984). Various possible influences caused by instrumental as well as geophysical effects were discussed and eliminated in the same study. Thus the main trend in the literature of the later studies attempted to find the ex-
planation of the mean observed downward vertical velocities in the mesosphere from radar measurements. Meanwhile, most of the later reported observed results obtained by radars agreed with this first study. The observational time and correspondingly averaging periods vary from several hours up to several days. The climatological studies on the vertical motion field are rarely available due to experimental costs and limitation of the capable instruments be able to provide convenient measurements. The mean observed downward velocities are close to 20-25 $\text{cms}^{-1}$ at mesopause altitudes. The first endeavour to explain the results
were connected to the possible observation of Stocke's drift, the difference between real Lagrangian and observed Eulerian motion, caused by propagated long-scale GWs in the mesosphere (Coy et al., 1986). Later this argument was discarded by Hall et al. (1992) revising the estimations of possible contribution in the downward effect caused by Stoke's drift in the upper mesosphere. Their estimations revealed values smaller than 4$\text{cms}^{-1}$. As the observed mean values of the vertical motion were several times larger, the stated effect was considered deficient for understanding the full picture of dissimilarity in the observed
and expected behaviour of the mean vertical motion in the polar summer mesopause.

Hoppe and Fritts (1995b) proposed a possible dynamical mechanism that caused biases on the radar measurements of vertical radial velocities. They found a correlation between mean vertical velocities and backscattered echo power, spectral width and the velocity uncertainties. The finding was connected to the influence of the GWs upward phases on the strength of the PMSE. The study argued that the presented mechanism should lead to biases on any radar measurements of the vertical velocities. A
similar effect was also found in the measurements at tropospheric altitudes (Nastrom and VanZandt, 1994).

Besides the direct observations, the estimations of the vertical wind from ground-based (Dowdy et al., 2001; Laskar et al., 2017) or satellite (Fauliot et al., 1997) measurements are also available, and are consistent with expectations.

Here we present the climatology of the 3D wind field retrieved from the PMSE 5 beam radial velocities measured during two summer seasons 2016 and 2017. The main focus in the study is on the estimation of statistical significant vertical wind
component, which is rather important for the mesospheric climatology at polar latitudes.

The paper is organised as follows: next chapter reviews the wind retrieval algorithm and a short description of the measurement technique. Main results and their discussion are available in the 3rd and 4th sections correspondingly, and we conclude our main findings at the end.





## 2 Measurements and wind analysis

During the summer months June to July 2016 and 2017 MAARSY was operated running a 5 beam experiment with a vertical beam and four oblique beams pointing North, South, East and West with a 10° off-zenith angle. The experiment was designed to obtain reliable 3D wind velocities from polar mesospheric summer echoes assuming that these echoes are an inert tracer. A

more detailed technical description of MAARSY can be found in Latteck et al. (2012).

The recorded raw voltages were analyzed with respect to the radial velocity, Signal-to-Noise ratio (SNR) and spectral width using the truncated Gaussian fitting (Kudeki et al., 1999; Sheth et al., 2006; Chau and Kudeki, 2006; Stober et al., 2018b). The fitting routine also includes an estimate of the statistical uncertainties for the radial velocities and spectral width. Further, we perform a $\chi$-square test to ensure that our assumptions of the fitted truncated Gaussian function holds. The optimal choice of

the number of incoherent integrations is done by the analysis routine itself, based on several thousands of manually quality controlled fits, and not fixed to avoid issues with potential experiment changes due to different measurement campaigns in both seasons. Thus, the number of incoherent integrations is maximized under the condition that the smallest spectral width within PMSE still can be resolved, which was determined for MAARSY to be in the order of 0.8 m/s.

The beam pointing was corrected for possible deviations from the nominal beam pointing direction applying Capon radar

imaging (Sommer et al., 2014, 2016). This corrections seem to be necessary to account for layering tilts and small scale structuring in the PMSE itself. Typically, the deviation from the nominal beam pointing is smaller than the MAARSY beam width of 3.6°. Similar to (Stober et al., 2018b) we also remove a possible contamination due to specular meteor echoes or meteor head echoes, before we start with the wind analysis.

In the wind analysis we put our focus on the proper assessment of the measurement uncertainties given that we want to

determine the 3D vector wind components with the best possible precision and accuracy. In particular, we want to address the long debate about the reliability of vertical wind measurements obtained from PMSE Hoppe and Fritts (1995b). Sommer et al. (2016) pointed out that PMSE appears to be more homogeneous using 10 minute averaging. We have used such time averaging to avoid issues related to localized scattering or aspect sensitivity. The vertical resolution of the wind retrieval is set to 500 m to account for the oblique beam sampling of the 300 m range resolution of the analyzed experiments.

Vector winds are obtained applying the wind retrieval applied in specular meteors outlined in Stober et al. (2017, 2018a). There is only one noticeable difference between the wind retrievals for meteor radars compared to the MAARSY. There is no need to include a correction for the full Earth geometry, if only small off-zenith angles are used in the observations. Briefly we describe the wind analysis with a particular emphasis of the non-linear error propagation. The start is the radial wind equation;

$$v_r(\theta,\phi) = u\cos(\phi)\sin(\theta) + v\sin(\phi)\sin(\theta) + w\cos(\theta) \ ,$$

here is $v_r$ the radial velocity, $u,v$ and $w$ are the zonal, meridional and vertical wind components and $\phi$ is azimuth angle given in mathematical coordinates (counterclockwise) with reference towards East and $\theta$ the off-zenith angle. Further, we determined the statistical uncertainty of the radial velocity from the spectral analysis $\Delta v_{rad}$. Assuming a Gaussian process, the total





statistical uncertainty of the radial wind equation can be written as;

$$dv_r = \left|\frac{dv_r}{du}\right| \cdot \Delta v + \left|\frac{dv_r}{dv}\right| \cdot \Delta v + \left|\frac{dv_r}{dw}\right| \cdot \Delta w + \left|\frac{dv_r}{d\phi}\right| \cdot \Delta \phi + \left|\frac{dv_r}{d\theta}\right| \cdot \Delta \theta \quad .$$

The $\Delta$ terms represent statistical uncertainties, which are a priori not all well-known. In particular, the statistical errors of the 3D winds are unknown and need to be estimated iteratively. The angular statistical uncertainties of $\phi$ and $\theta$ are determined to

be $2°$ based on heuristic arguments.

In addition, we also apply a regularization in space and time by computing the vertical and temporal wind shear for each vector component. This is done by binning the data in space and time using a reference time and altitude grid. The vertical shear is computed from the bin above and below. A similar procedure is applied in time. If there is no measurement in one bin available due to a vanishing PMSE or at the upper and lower edges of the PMSE layer, we just use the difference between the reference

bin and the available neighbors. This regularization can be considered as space-time Laplace filter.

There is another aspect that is worth to be considered related to the vertical and temporal total wind shear. The total wind shear is a good measure to account for the irregular sampling of the data compared to our reference binning. Typically, the measurement of a radial velocity did not occur exactly at our reference time and altitude for each bin. Further, the shear terms provide a physical constraint of how variable the wind field might be for a certain time and altitude, adding valuable information

about the geophysical uncertainty of the observations. Each measurement within a time altitude bin is also weighted using the following expressions;

$$\Delta_{tshear} = A_{tshear} - A_{tshear} \exp{-\frac{(t_m - t_{ref})^2}{(0.5 \cdot dt_{ave})^2}}$$

and for the altitude

$$\Delta_{ashear} = A_{ashear} - A_{ashear} \exp{-\frac{(alt_m - alt_{ref})^2}{dh_{ave}^2}} \quad .$$

Here are $A_{ashear}$ and $A_{tshear}$ the vertical and temporal shear amplitude, $alt_m$ and $t_m$ are the altitude and time of the measurement, $alt_{ref}$ and $t_{ref}$ are the reference time and altitude for a bin and $dt_{ave}$ and $dh_{ave}$ are the averaging kernels provided by the user and should be in the order of the bin widths or if oversampling is applied larger than the bin size. In this study we used an oversampling in time $dt_{ave}$ of 20 min and a temporal resolution of 10 min. In the vertical coordinate no oversampling is applied.

The described retrieval solves the radial wind equation using all the weights listed above iteratively. The initial guess is obtained using a least square weighted by the statistical uncertainty of the radial velocity measurement and a shear amplitude of $5\,\mathrm{ms}^{-1}$ for the temporal and vertical shear, respectively. The other weights are set to zero for the initial guess. After each iteration all weights are updated and a new solution is computed until the result doesn't show any significant change compared to the previous one. Typically 3-5 iterations are required. So far we have not set a fixed threshold for convergence.

We validated the described algorithm making use of tropospheric measurements of zonal and meridional winds gathered during a campaign in February and March 2016 with the MAARSY radar and compared these with the ECMWF (European Centre for Medium-range Weather Forecasts) data. The comparison is based on hourly data and a one kilometer vertical resolution. The





correlation is as high as 0.987 for the zonal and 0.994 for the meridional wind component using all altitudes. The comparison indicates that the derived wind retrieval should be suitable to obtain reliable winds in the mesosphere as well.

## 3 Results

The occurrence of PMSE usually increases after mid of May, reaches relatively high visibility at the beginning of June and decreases after the end of July fading out in the second half of August (Rapp and Lübken, 2004; Latteck et al., 2008). We analysed PMSE observations with MAARSY between 1st June and 31st July of the years 2016 and 2017 and considered this time interval as a summer season in this paper. The observations are almost continuous, excluding the gap of several days caused by a change in the radar experiment settings. The total data consists of to 49 days in 2016 and 57 days in 2017.

The aim of this study is mainly connected to vertical motion field, the mean climatology of the vertical wind velocities and the relevance of using PMSE as an inert traces to obtain neutral wind velocities. Therefore, we present the mean seasonal vertical velocities as a function of altitude, plotted on the left side of the Figure 1. Considering the velocity axis directed upward, negative values of the vertical velocity represent the downward motion, and positive corresponds to upward motion.

Weighted averages (solid red line) are characterized with different behaviour in three sections for both seasons. The middle, vertically almost invariable values between 84-87 kilometre altitudes and the two outer sections with sharp decrease/increase at the lower/upper edges of the PMSE observable layer.

Weighted averages are presented as $\overline{w} = \Sigma w_i x_i / \Sigma x_i$, where weights are $x_i = 1/\Delta w^2$ and corresponding uncertainties of the averages will be $\sigma_{avg} = 1/\sqrt{\Sigma x_i}$.

To ensure a statistical significant results, we filter the velocity measurements by PMSE occurrence. We averaged only those data at heights corresponding to the PMSE occurrence rates more than 20%. The remaining data results in average PMSE-layer between 81.5-89.5 km altitudes in 2016 and 82-89 km in 2017 considering the indicated threshold. PMSE occurrence rate at a given altitude is a ratio of available data with PMSE, defined by a Signal-to-Noise Ratio (SNR) greater than -8 dB, to the full length of the observation time. We use this definition here and on following plots as well. The weighted average values for the whole altitude range are -10 cms$^{-1}$ and -7 cms$^{-1}$ for the summers of 2016 and 2017 respectively. The simple mean values of observed vertical velocities, presented in the previous studies, were close to -20 cms$^{-1}$ (Balsley and Riddle, 1984; Meek and Manson, 1989; Fritts et al., 1990). We have also obtained similar values applying simple averaging 19 cms$^{-1}$ downward for both observed years (dark blue curve on the figures).

However, due to the full error propagation, it is possible to weight all measurements with its corresponding statistical uncertainties, which is adequate as larger velocities are often associated with larger errors and should be weighted correspondingly. Thus, the further discussion and presented results are weighted averages only. The terms "mean" or "average" regarding the wind velocities should be understood as weighted averages.

Mean vertical velocities on Figure 1 at altitudes above 87 km are strongly upward, while below 84 km they are downward with relatively high magnitudes, but closer to zero values in between. Hence, it is more consistent to discuss each of the altitude sections separately. In particular, as different physical processes related to the microphysics of the tracer are involved. We turn



back to this in detail in the next section.

In spite of the seasonal mean distribution of the vertical velocities, considering the consistency of short time averages to the seasonal climatology is relevant. Daily distributions of the occurrence rates of the PMSE throughout the summer season and the altitude range between 78 and 92 km shown on the upper panels of Figure 2. The spatial distribution of the observed PMSE,

as well as the seasonal occurrence, is in agreement with previous observations from the same location (Hoffmann et al., 1999; Latteck et al., 2008), which shows that the analysed two summer seasons are representative for a mean climatology.

The second panels of the Figure 2 show the colour-coded daily means of the observed vertical velocities corresponding to the occurrence of PMSE greater than 20%. The main trend of the daily mean observed vertical motion below 87 km altitude is downward (negative values) and above – upward (positive values). However, there are days with predominantly upward (e.g.

the day of the year (DoY)=173 in 2016 or DoY=183 in 2017) motion within the discussed altitude range. This contradicts the conclusion in the previous study by Hoppe and Fritts (1995b) that any mean vertical velocity observed with PMSE should be biased toward downward behaviour. The tendency of relatively strong downward shape below 84 km (dark blue colours) and weaker, close to zero values between 84-87 km altitudes (indicated as whitish) are also remarkable. In this way, the summer behaviour of the vertical velocity mainly corresponds to the mean seasonal profile shown on Figure 1.

Seasonal distribution of daily mean values of zonal and meridional velocities at a given altitude is plotted on the last two panels of Figure 2, correspondingly. Shown summer climatology for both horizontal components agree with those observed previously using meteor radars (Dowdy et al., 2007; Jacobi, 2011; Wilhelm et al., 2017). In particular, the zonal wind shows a well-defined westward directed (here, negative values) and increasing in magnitude at the lower altitudes. The meridional wind component tends to be equator-ward (also negative values) but is weaker in magnitude than the zonal one. An expected

tendency of seasonal behaviour is outlined, on the whole, however the zonal wind reversal at the PMSE upper altitudes is rarely observed.

Figure 3 shows diurnal values of PMSE occurrence (upper panels) and PMSE velocity components – vertical, zonal and meridional respectively on the second, third and lower panels of the figure, but averaged as a function of universal time (UT) and altitude. The half-hour temporal bins, with 10-minute time-step, are used to average all available velocity components at a

given altitude.

We calculate the mean diurnal occurrence of the PMSE in the same way but looking to the ratio of all available PMSE (SNR>-8$dB$) within the given 30 minutes of each observed day to the full length of the measurements. The results plotted on the upper panel on the Figure 3. The black contour line indicates the PMSE occurrence equal to 20%.

A significant increase in PMSE occurrence is observed during UT 12:00 – 16:00 and a secondary peak is also noticeable in

the early morning hours for both years. The similar behaviour of the PMSE occurrence was also reported in previous studies and was connected to the semidiurnal tidal variations (Hoffmann et al., 1999) but the diurnal production/reduction of electron densities is important too. Semidiurnal tidal behaviour is also significant in the horizontal wind components shown on the lower two panels of Figure 3. However, the vertical velocity component (second panel) does not show any clear dependence neither on the UT period nor on the PMSE strength as well. There are weak upward behaviour highlighted shortly at 6:00 UT

for 2016 and after 9:00 UT for 2017 around 85 km altitude, but the velocity values are mainly in good agreement with the




mean seasonal profile.

We examined the mean vertical velocity dependence on PMSE strength as well as on the universal time. The results show that the mean value of the vertical velocity and its altitude behaviour is not dependent on both mentioned parameters. Second and last columns of Figure 4 show 2D histograms as a function of the PMSE/SNR and vertical wind velocity for years 2016 and
2017 respectively. Gaussian-like distribution is symmetrical for most of the PMSE strength values except some extreme cases where they are statistically sparse. The mean vertical profiles of vertical velocities shown on the left panels to the histograms correspond to the three separate intervals of PMSE strength. The lower correspond to PMSE with SNR larger than -8 dB but lower than 5 dB; middle plots are done for velocities corresponding PMSE/SNR between 5 and 20 dB and the upper plots - to PMSE/SNR more than 20 dB. The behaviour of mean vertical velocity profile averaged at high PMSE values for summer 2017
is unlike to all others because of lack of the relatively strong PMSEs during summer 2017. The difference is insignificant for the all other cases. No correlation between mean vertical velocity and PMSE strength is found, unlike previous reports based on several hours of measurements.

## 4   Discussion

Outcomes from a given tracer indeed depend on its occurrence. Results obtained from a tracer are acceptable if it appears randomly but not in certain circumstances associated with a target parameter. The discrepancy of vertical velocity measurements in PMSE compared to the predictions from global circulation models seemed to have been explained by Hoppe and Fritts (1995b) who argued the influence of background dynamics on appearance of PMSEs resulted in a bias on all measurements of vertical motion. We discuss presented mechanism first and return to our findings later.

Hoppe and Fritts (1995b) proposed a mechanism for the observed bias on the mean downward PMSE vertical velocity based on analysis of the five hours PMSE (EISCAT - European Incoherent SCATter) radar measurements of the vertical beam radial velocity. They found that upward velocities preserved more than 10 minutes led to increasing the variance of the motion field and tended the weaken PMSE SNR. They concluded that downward the biases in PMSE mean vertical velocities is due to GW increasing the turbulence after certain (upward) phase of the vertical motion field and made PMSE appearing weaker. The
mean downward vertical velocities therefore correlated to the larger measurement uncertainties and variances in the low-pass filtered vertical motion field.

Large spectral width is usually an indicator of the strength of turbulence. The increase of variances of the radar Doppler velocities and measurement uncertainties is expected while observing the stronger turbulent air. The weakening of PMSE at the large spectral width conditions is also known from the literature (Stober et al., 2018b; Chau et al., 2018). Hence, the explained
mechanism of downward bias in PMSE vertical velocity is realistic. The question that remains is on how systematic the described process is, and how far it affects the seasonal climatology.

We investigated the correlation of the retrieved mean vertical wind component on their statistical uncertainties for the summer seasons of both discussed years. The available data allows us to repeat only such correlation because of the difference in time




resolution of datasets. The results are shown on the Figure 5. The histograms on the upper panels represent the distribution of measurement uncertainties $\Delta w$. The blue circles on the lower plots correspond to the vertical wind mean values for only those measurements which uncertainties are in the range marked out by horizontal solid lines. Each "sections" of uncertainties corresponds to the relatively same amount of data-points. The red circles show mean vertical values corresponding to all those

uncertainties lower than a given point. Therefore, Figure 5 is the analogy of the Figure 4 in Hoppe and Fritts (1995b).

The result shows the opposite to those found by Hoppe and Fritts (1995b). Later we examined the resembling correlations concerning each available 6-hour sections. Only for a few cases, as shown by Hoppe and Fritts (1995b), with a positive correlation between mean downward values and uncertainties, were found. Figure 6 shows such examples for both years. Following our results, we can summarise that the comprehensible dynamical process obtained from the relatively short period of observations

can represent the localised event, but is might not be representative to explain the climatology of vertical motion observed in PMSE.

The time resolution of our dataset does not allow us to reexamine the mean vertical velocities as a function of variances on short-time scale motions. Nevertheless, to enforce our argument, we refer to the Figure 4. No clear evidence of the dependence of vertical velocity distribution on PMSE strength available from the seasonal 2D histograms. However, the results of the mean

vertical profiles of the vertical velocity corresponding to different strength range of PMSE are almost equal. The magnitude of mean downward values slightly change at lower altitudes and middle strength of PMSE, but does not at the centroid heights (84-87 km) of the PMSE occurrence, except those for SNR>20 dB during the year 2017. This particular case is due to the small amount of the data-points. In general, the PMSE in 2017 characterises weaker occurrence than during 2016 (upper pannels of Figure 2 and Figure 3). We show that the seasonal climatology of the mean vertical velocities retrieved from PMSE is not an

evident function of the PMSE Strength.

Figure 1 clearly shows the three well-developed sections of the mean vertical profile of the vertical velocity at PMSE altitudes. The central sections are downward but close to the zero values with mean equal to -7 and -2 cms$^{-1}$ in 2016 and 2017 respectively between 84-87 km altitudes. Thus discussing the influence of sedimentation speed of ice particles on the measurements of PMSE radial velocities become substantial. The appearance of ice crystals with a charge attached to them is determinative

to slow down electron diffusivity and create radar backscatter in the presence of turbulence. Further, the ice particles are much heavier than the ambient atmosphere, and they sediment down due to the gravity. Such motion affects the vertical velocity measurements eventually. According to the present knowledge of the mesospheric ice physics, those particles created at PMSE altitudes where the atmospheric temperature can reach below the frost point (Lübken and von Zahn, 1991; Lübken, 1999; Rapp and Lübken, 2004) for the typical mixing ratios of the water vapour (Garcia, 1989).

Ice particles are sedimenting slowly downward. The sedimentation velocity depends on size and shape of the particles as well as the ambient atmospheric density. We now discuss the possibility to use PMSE as an inert tracer to obtain realistic vertical velocities in the summer mesosphere but not pretending the quantitative analysis here. Hence, we take into account the idealised case, when ice crystals have a spherical form. Following Reid (1975) the falling velocity of such ice particles is:

$$v_{ice} = \frac{\rho g r}{2n} \left( \frac{\pi}{2mkT} \right)^{\frac{1}{2}}$$  (1)





Where $\rho$ is the density, and $r$ is the radii of the falling sphere, $g$ is the acceleration due to gravity, $n$ is the number density and $T$ - absolute temperature of the air, $k$ - Boltzmann's constant, $m$ is a mass of the air molecules. Mean summer air number density and temperature profiles are taken from NRL-MSISE00 model (Picone et al., 2002). The mean mass of the molecule is calculated as a weighted mass based on the densities of main constituents ([O], [O2], [N2]) at the mesospheric altitudes.

We use $910 kgm^{-3}$ as the ice mass density, $\rho$. We consider two scenarios of the radii $r$ distribution. One, following (Berger and Lübken, 2015) who modelled the ice formation and growing to derive the expected particle sizes at different altitudes. It results from 5 to 35 nm ice spheres from 90 down to 82 km altitude range and a rapid decrease to a few nanometers at the lower altitudes. Observations sometimes show the NLC with the ice particles in 100 nm (von Cossart et al., 1999). Therefore the second scenario considers the extreme size distribution of ice grains from 10 to 100 nm at the above mentioned altitudes.

The results are plotted in light blue colour on the Figure 1 (all panels) as the solid and dashed lines corresponding to the first and second scenarios. Green solid and dashed lines on the same plots represent the corrected observed mean velocities (red curve) after eliminating the influence by ice falling speed (e.g. $\overline{w} - v_{ice}$). The resulted velocities became upward for both years after excluding the effect from the ice descend with expected magnitudes in range of few $cms^{-1}$ at the discussing altitudes of high PMSE occurrence (84-87 km). We argue that this result can be understood as a clear evidence of direct observation of the

mean upwelling in the summer mesosphere. Also PMSE can be discussed as an inert tracer for the vertical neutral wind but only at certain altitudes.

We should notice that parameters of the ambient atmosphere taken from the NRL-MSISE00 model corresponds to summer 2016 and applied for both discussed years due to the availability of model outputs.

In the previous section, we have proposed the relevance of discussing the observed/retrieved vertical velocities separately at

the middle and outer altitudes of PMSE occurrence. The evidence of the sharp increase/decrease of the summer mean vertical velocity at the edges above 87km and below 84 km on the Figure 1 should have a separate explanation.

The vertical motion activity in the region as well as in the atmosphere can be a result of wave propagation, created turbulence due to shear instability or after breaking of GWs (Hecht et al., 2000). Such disturbances indeed transport air masses on the corresponding scales. Relatively large-scale motions are usually accompanied with short-scale wave processes. The effect of

the ambient air motion on the PMSE are widely observed and can be explained as following of ice clouds to the atmospheric behaviour.

A typical day as an example of PMSE temporal behaviour is shown on the Figure 7. Static ground-based measurements are not capable to observe intrinsic motion but look in the specific fixed volume in the atmosphere. Hence, the horizontal transport of tracer is also essential to describe the observed vertical fluctuations. The large-scale GWs generated and transported from the

lower altitudes have a characteristic vertical wavelength of 15-20 km at upper mesospheric altitudes (Mitchell and Howells, 1998; Schmidt et al., 2018). Up- or downwelling phase of such motion exceeds the width of PMSE layer and are capable to transport air masses from lower or upper heights where the ice clouds and PMSE are not present.

We plot the idealised sinusoidal wave sketch on Figure 7 as the black dashed line. The beginning phase at 01:00 – 05:00 UT corresponds to the downwelling and PMSE structures follow the air to the lower altitudes. There they disappear due to melting,

sublimation, or horizontal advection moves in air masses without PMSE. The measured Doppler velocities are only available as



long as the radar detects an echo. Uprising phase of a vertical component of propagating wave starts after 05:00 UT (on given example) and as air is depleted of PMSE at lower altitudes the evidence of upward vertical velocities are underrepresented in the measurements. Following the progress in time and space, horizontal transport plays a role again and contributes the PMSE structures in the radar field of view. The vertical velocities are available again on the central altitudes of PMSE layer.

The similar but opposite picture in respect of motion direction is evident at the upper edge (close and after 10:00 UT on Figure 7). Finally, the number of data points with downward motions are dominant at the lower altitudes, and correspondingly there is an excess of upward velocity measurements at the upper/top heights. The effect is more evident when the distance from the PMSE central altitudes is considered. Here we proposed a geophysical explanation of the sharp trends at the edges of the vertical profile of the mean vertical velocity retrieved from five-beam PMSE Doppler velocities.

**5 Conclusions**

We present a climatology of PMSE mean vertical velocities at polar mesospheric altitudes for two summer seasons. The main objective of this study was a detailed analysis and understanding of these velocities, since previous studies have shown that they are not in agreement with the expected upwelling at the MLT due to the residual circulation.

Our results indicate that continuous 3D wind measurements are necessary to derive a mean vertical motion in the MLT from
PMSE observations. PMSE can be considered as an inert, but not as a passive tracer. As a result reliable climatologies are only obtained at altitudes with a 24-hour PMSE coverage. Further, we have to consider, that the charged aerosols inside the PMSE reduce the electron diffusivity and are basically sedimenting with the ice particles. Taking into account such expected sedimentation speeds, the weighted mean vertical velocity in the central altitudinal region of PMSE (i.e., 84-87 km) shows a small upward motion. Such upward motion is in agreement to what predicted and expected for the polar summer mesosphere.

However, at the upper and lower altitudinal edges of PMSE, the weighted mean vertical velocities show unexpected behavior. They tend to be up/down-ward at upper/lower edges. We proposed that this behavior is due to the non-uniform appearance of PMSEs in altitude and time. For example, an upward velocity is not picked up by the measurements at the lower altitudes, since no PMSE can be advected from below the PMSE layer, while during downward motion PMSE is basically advected from the altitudes above and the tracer disappears merely after melting the remaining ice clouds. A reversed effect occurs on the upper
edge of the layer. Horizontal advection of PMSE plays an important role at the middle altitudes where's the ratio of up and downward measurements becomes independent on the phase of vertical motion field.

We also examined the possible biases on the PMSE vertical velocity measurements from certain phases of short-period gravity waves earlier proposed by Hoppe and Fritts (1995b). Based on the statistical analysis of the available long dataset we found that the mean behaviour of PMSE vertical velocity does not depend on the PMSE brightness at altitudes between 84-87 km, where
we found a sufficient high occurrence rate. No positive correlation between downward velocities and velocity uncertainties were found as well. We argue that the results of short-time measurements cannot be extrapolated to a climatological behaviour.





*Data availability.*  Contact Gunter Stober for MAARSY PMSE wind data (stober@iap-kborn.de). Detailed information regarding access to ECMWF data can be found at http://www.ecmwf.int.

*Author contributions.*  All authors contributed to the editing of the paper. N. Gudadze provided the data analysis. The raw data was processed by G. Stober and is available upon request. The work is supervised by Prof. J.L. Chau.

5   *Competing interests.*  The authors declare that they have no conflict of interest

*Acknowledgements.*  This work was supported by the WATILA Project (SAW-2015-IAP-1).



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



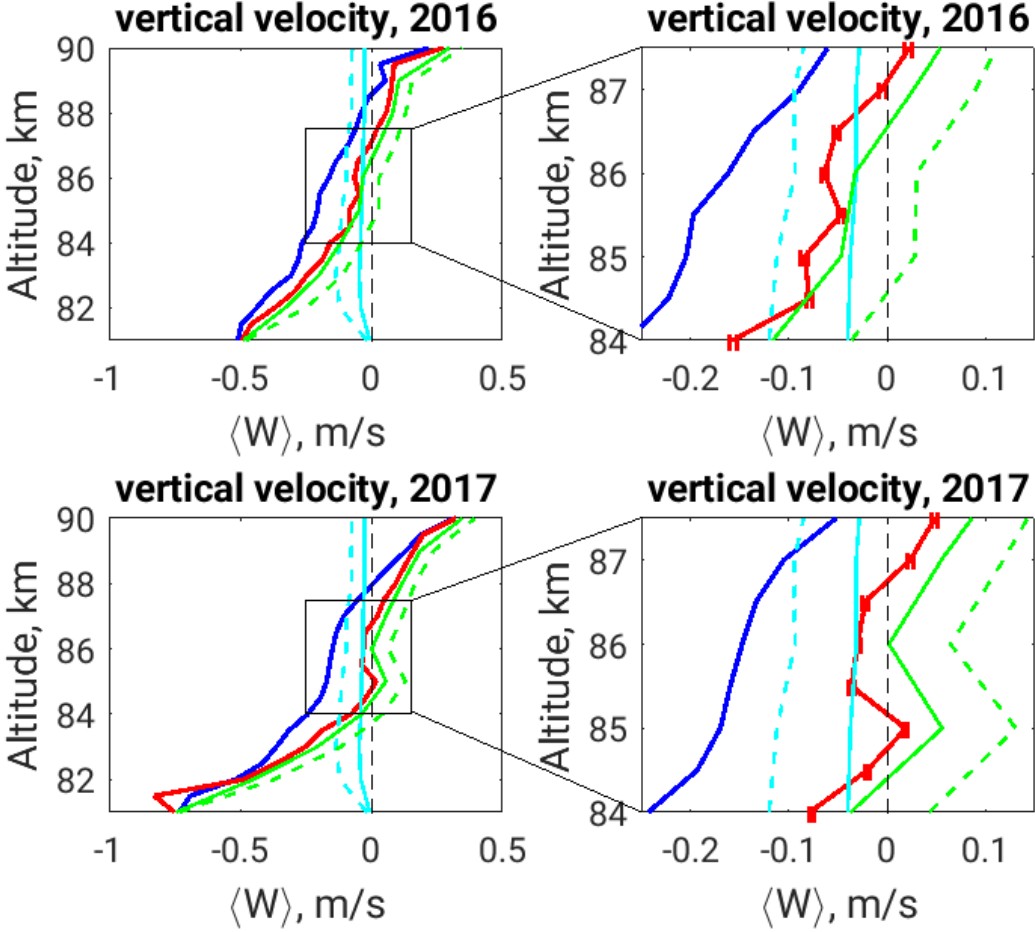

**Figure 1.** Mean summer profiles for (blue) mean and (red) weighted mean vertical wind velocities; (dashed and full light blue) sedimentation speed of ice spheres for two scenario of their size distribution; (dashed and full green) results after eliminating of ice sedimentation from weighted mean profiles. Right panels show increased version of the altitude range in black squares at the left panels




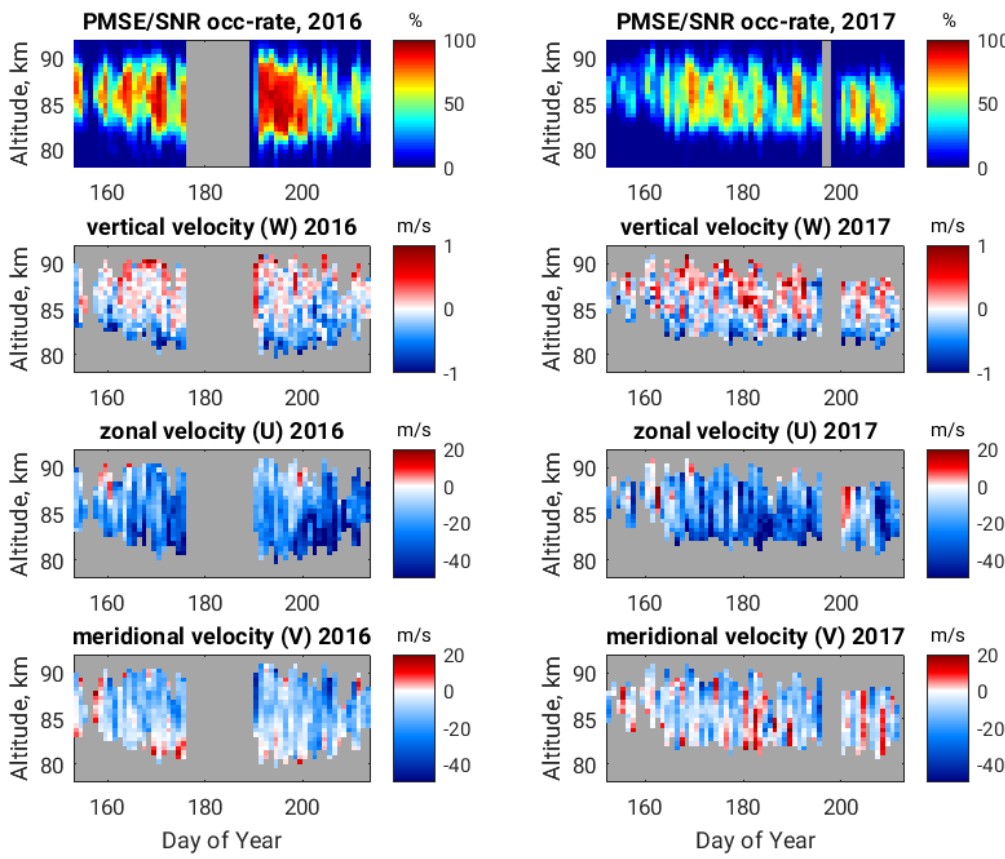

**Figure 2.** Seasonal distribution of PMSE/SNR diurnal occurrence rates (first row) and weighted mean diurnal profiles of vertical, zonal and meridional velocity vector components given on following rows. Positive wind values corresponds up-, east- and north-ward directions respectively. Wind components correspond to those PMSE occurrence greater than 20%. Grey color indicates no data background.





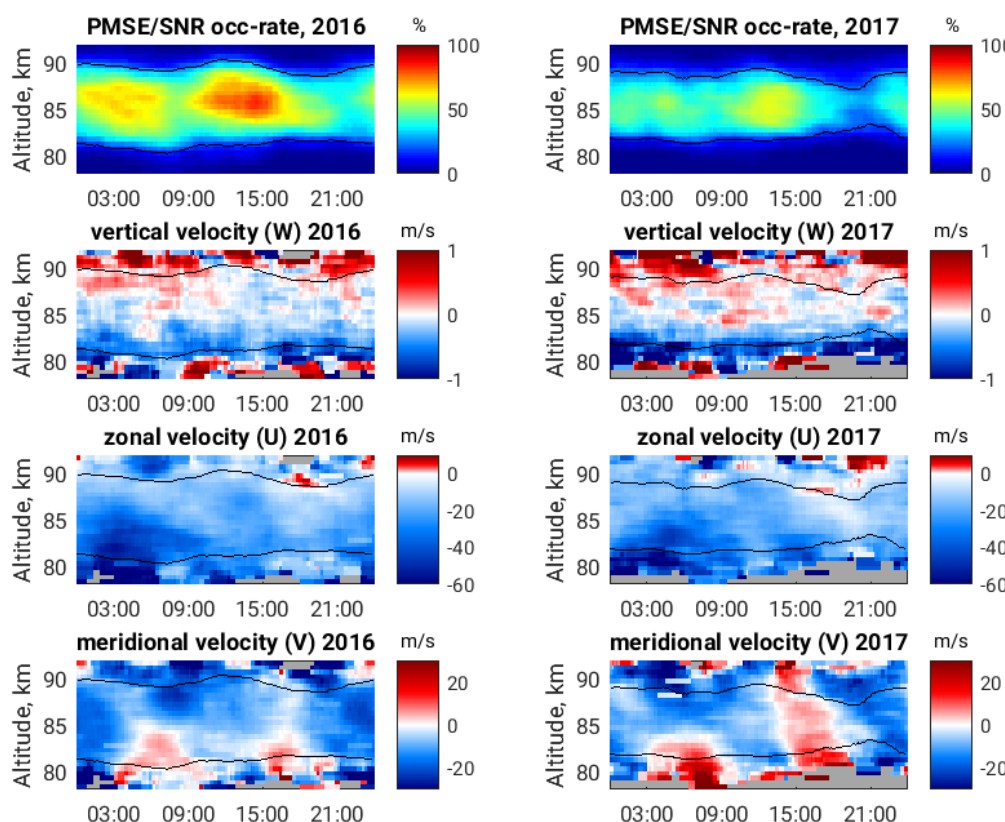

**Figure 3.** Universal Time (UT) Dependence of PMSE/SNR occurrence rate (first row) and weighted mean diurnal profiles of vertical, zonal and meridional velocity vector components given on following rows. Positive wind values corresponds up-, east- and north-ward directions respectively. Black solid contour lines correspond to 20% PMSE/SNR occurrence rate; Grey color indicates no data.





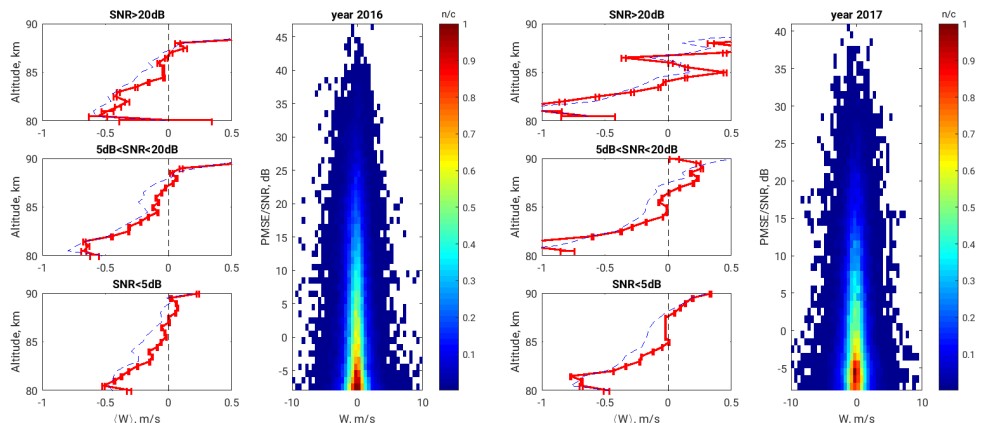

**Figure 4.** Vertical profiles of mean vertical velocities corresponding to given range of PMSE/SNR (first and third columns for summers of 2016 and 2107 respectively). 2D histogram of vertical wind measurements as a function of PMSE SNR (colorcoded panels)

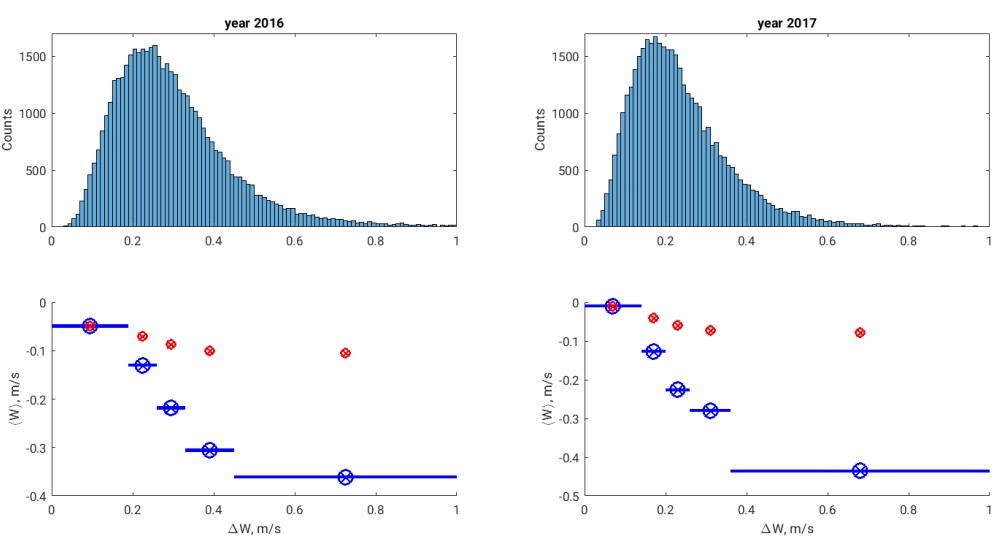

**Figure 5.** Upper panels - histograms of vertical velocity uncertainties; Lower panels - mean vertical velocity as a function of their uncertainties. Blue circles represent mean velocity values corresponding to velocity uncertainties range indicated with blue horizontal line on them. Ranges are divided equally in respect of similar datapoints for averaging. Red circles are mean vertical velocities for all those uncertainties lower than given point



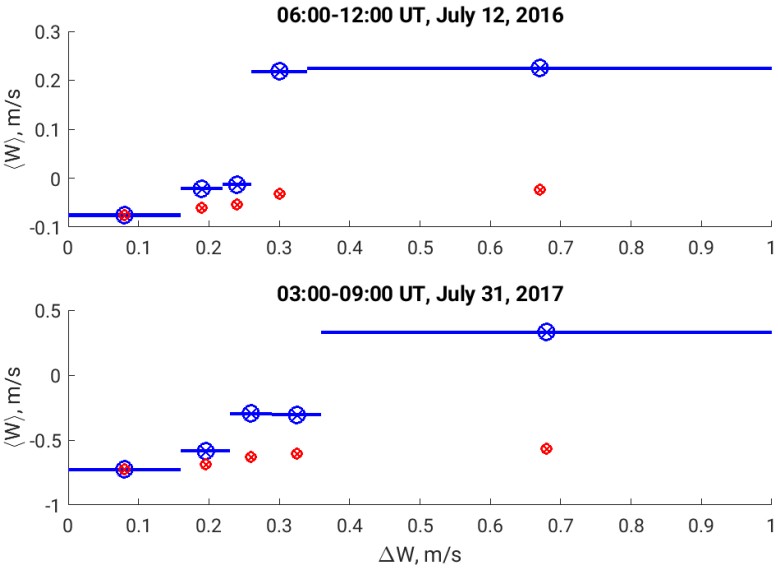

**Figure 6.** Couple of examples of positive correlation between mean vertical velocities and their uncertainties as shown by Hoppe and Fritts (1995b). The exact time intervals are indicated on the plots. Further details can be found in the text.

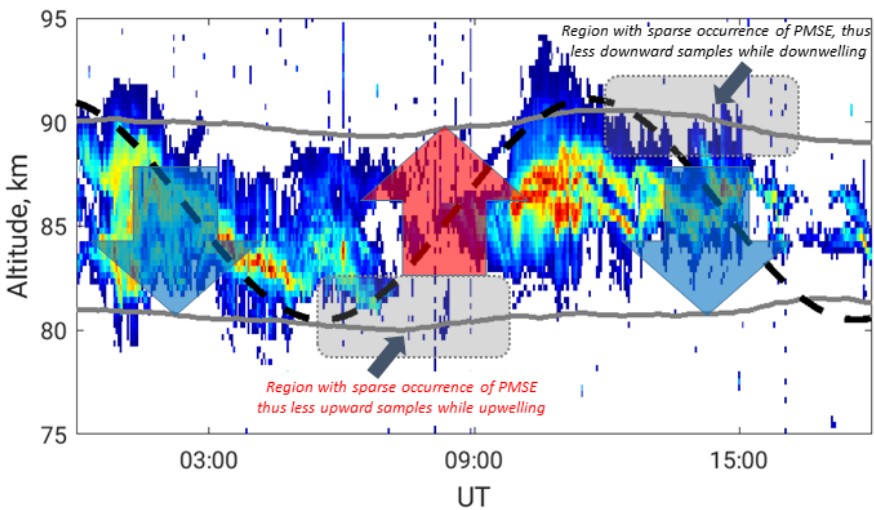

**Figure 7.** Example of typical PMSE structure observed on 29th of July, 2016 and sketch of idealised propagated sinusoidal wave at PMSE altitudes. Arrows indicate the up or downward phase of the wave. Colour coded PMSE is thresholded by SNR=-8 dB. Thus, white background corresponds to no data. Grey line show boundaries of seasonal average of diurnal PMSE occurrence more than 20%. Sketch highlights the sampling issues of PMSE and velocity measurements on the edges respectively.