# Peer review of "Can VHF radars at polar latitudes measure mean vertical winds in the presence of PMSE?"

_Atmospheric Chemistry and Physics, 2018_

## Referee Comment (RC1) · Anonymous Referee #1 · 5 Nov 2018

General Comments

The paper addresses the issue of whether radars can measure the vertical wind in the polar summer MLT after compensating for the fall speeds of the ice particles responsible for the radar backscatter.

While the paper is basically OK, it reads as if it is addressed to those who already know about the issues involved. A small increase in the length of the paper to provide more background material could lead to a broader readership. For example, measurements are reported for two summer seasons using the so-called MAARSY radar located in Andoya, Norway. Nowhere, however, is the MAARSY radar described, including the operating frequency. The reader should not have to go searching for such basic details.

[Figure]

The title of the paper could also be better expressed. Given that the aim of the paper is to show how the true vertical wind can be measured in the presence of the downward motions of the ice particles that dominate the radar backscatter, maybe the title might be better framed as "Can VHF radars at polar latitudes measure mean vertical winds in the presence of PMSE?" In that sense, the first two sentences in the Conclusions best encapsulate the motivations for the paper and it would help the reader if they were repeated up front in the Introduction.

The paper starts (p2) with an incorrect discussion of how the zonal wind structure in the MLT reverses sign. It does not occur through the gravity wave momentum deposition causing a direct "drag" on the zonal winds, with the meridional flow arriving as incidental by-product. It is just the opposite. 1. The eastward (westward) momentum deposition in the summer (winter) MLT drives a meridional summer-to-winter pole circulation. 2. This leads to rising (sinking) motions over the summer (winter) pole with consequential adiabatic cooling (heating). 3. Hence, the zonal-mean latitudinal temperature gradient in the MLT is reversed relative to that in the stratosphere and the zonal-mean zonal winds in the mesosphere change sign through the thermal wind relation. This section should either be modified appropriately or deleted entirely to avoid confusion.

Confusion can also occur through mixing the use of vertical motions and vertical winds. For example, in the caption to Figure 1 the blue and red curves are labelled as "weighted mean vertical wind velocities" when they are actually vertical motions strongly weighted by the sedimentation speeds of the ice particles, i.e. not the the vertical velocity of the neutral atmosphere. It is recommended that the terminology throughout the paper be corrected to ensure there is no misunderstanding of what is a vertical motion and what is a wind.

It is realised that English is probably not the first language of the authors, but the text needs considerable editing and proofreading to improve the readability. Definite articles such as 'the' are often used incorrectly, which sometimes makes reading and interpretation of the contents difficult.

Specific Comments

P1, L12 "Disappearance" not "disappearing"?

P3, L15 Stoke's drift.

L4, L21 Brackets required around Hoppe and Fritts, 1995b

P5, L27 I do not understand "vertical shear amplitude of 5 m/s". Should this not be 5 s-1?

P8, L22 I do not understand the sentence starting "They found..." Is "continued' meant, rather than "preserved"?

P9, L5 "threshold" rather than "point"?

P9, 14 "is" before "available".

P9, L25 "reduced" rather than "slow down".

P9, L26 "sediment due to gravity"

P9, L32 "Presenting", not "pretending".

P10, L5 Remove brackets around Berger and Lübken, 2015.

P10, L34 "downward motions" not "the downwelling"?

Figs. 2, 3 "Positive wind values correspond to"

Fig. 4 Do the 2D histograms represent "vertical wind measurements" or 'vertical speed measurements?"

What do the dashed blue lines in Fig 4 represent?

---

## Referee Comment (RC2) · Anonymous Referee #2 · 15 Nov 2018

The authors use observations by the MAARSY radar at Andenes, Norway, to investigate the vertical wind velocities and to shed light on the problem of the downward bias. The observations are done through reflections at mesospheric cloud particles, and applying a newly developed wind analysis method. If ice sedimentation is taken into account, an upward mean flow is found, qualitatively consistent with modelling results. An upward/downward bias at the upper/lower edge of the measuring volume is observed, which is explained by wave motion and uneven sampling connected with melting of ice at lower altitudes. The authors also analysed possible dependence of vertical winds on PMSE brightness and wind uncertainty. While the results do not completely support earlier observations at shorter time scales, the authors conclude that the results of short-time measurements cannot be extrapolated to a climatological

behaviour.

The observations and results are interesting and relevant, and the paper should eventually be published after some revision.

The paper needs a thorough revision of English language use. While most of the text is understandable, sentence structure and grammar is partly not acceptable.

The description of the measurements is insufficient for readers not familiar with the topic. More details, and even basics of the MAARSY radar; e.g., the geographic coordinates are only provided in the abstract. More information is necessary here. Few more information on the experiments would also be helpful. Have the experiments been run continuously during both seasons? Which is the grid size for binning (page 2, line 5)? This and more information would improve the paper.

Specific issues:

P2 l 17: "...considering the Bousinesq approximation..." delete this part of the sentence. Continuity equation does not require incompressible fluid.

P3, ll 22-24: The mechanism is described later, but should be outlined here.

P3, l 28: This sentence requires information that is only provided below or missing, namely the latitude of the observations, and an explanation what "PMSE 5 beam radial velocity" means.

P4, l 2: What does MAARSY stands for? It should not only be provided in the abstract.

P4, l 22: more homogeneous than what?

P7 l 17: Jacobi, 2011 shows midlatitude winds, not polar ones. If you want to add more references, you may wish to refer to radar based wind climatologies like Portnyagin et al., 2004.

P8, l 16: ".. but not in certain circumstances associated with a target parameter." I do

not understand what this means.

P9 l5 /Figures 5 and 6: I did not understand what the red points mean.

Figure 1: The blow-up on the right panels is not necessary in my opinion. The vertical resolution is 500 m and the effect is visible on the left panels also.

Minor comments

P2, l 29: admitted -> considered

P6, l 7: excluding the -> except for a

P6 l 25: "We have also. . ." something went wrong with this sentence

P8, l 22: downward the -> the downward

P9, l 4: remove "relatively"

P 16, l 28: Stober et al., 2018: refer to the final revised paper

Reference

Portnyagin, Yu., et al., 2004: Mesosphere/lower thermosphere prevailing wind model. Adv. Space Res., 34, 1755-1762, https://doi.org/10.1016/j.asr.2003.04.058.

―――――――――――――――――――

---

## Author Comment (AC1) · 24 Jan 2019

**Authors response to the reviewer#1 on "Are mean vertical velocities from PMSE a good representation of mean vertical winds?" by Nikoloz Gudadze et al.**

We thank the referee for her/his helpful feedbacks and careful reading of our manuscript. We appreciate all the efforts improving the manuscript and acknowledge the given comments and suggestions. We provide detailed replies to all the raised comments listed below. All changes in the manuscript will be highlighted as bold text.

The remarked language changes are also included.

**General Comments**

**Comment:**
**"For example, measurements are reported for two summer seasons using the so-called MAARSY radar located in Andoya, Norway. Nowhere, however, is the MAARSY radar described, including the operating frequency."**
> Reply:
> *A paragraph with a general description of the radar (MAARSY) is added in the second section (Measurements and wind analysis)*

**Comment:**
**The title of the paper could also be better expressed. Given that the aim of the paper is to show how the true vertical wind can be measured in the presence of the downward motions of the ice particles that dominate the radar backscatter, maybe the title might be better framed as "Can VHF radars at polar latitudes measure mean vertical winds in the presence of PMSE?" In that sense, the first two sentences in the Conclusions best encapsulate the motivations for the paper and it would help the reader if they were repeated up front in the Introduction.**
> Reply:
> *We agree to reviewer suggestion and changed the title.*

**Comment:**
**The paper starts (p2) with an incorrect discussion of how the zonal wind structure in the MLT reverses sign. It does not occur through the gravity wave momentum deposition causing a direct "drag" on the zonal winds, with the meridional flow arriving as incidental by-product. It is just the opposite. 1. The eastward (westward) momentum deposition in the summer (winter) MLT drives a meridional summer-to-winter pole circulation. 2. This leads to rising (sinking) motions over the summer (winter) pole with consequential adiabatic cooling (heating). 3. Hence, the zonal-mean latitudinal temperature gradient in the MLT is reversed relative to that in the stratosphere and the zonal-mean zonal winds in the mesosphere**

change sign through the thermal wind relation. This section should either be modified appropriately or deleted entirely to avoid confusion.

Reply:

*Regarding the comment on zonal mean zonal wind behaviour according to thermal wind relation, we totally agree to the critics. However, cold summer mesopause and therefore zonal wind vertical behaviour is a byproduct of the chain process induced gravity wave momentum deposition (or zonal wave drag). It's accepted, that the sentence "Such forcing decelerates the westward zonal wind on the corresponding heights and causes widely observed wind reversal at the lowest thermospheric altitudes" (p.2 l.14) can be understood as a direct effect. We have modified it. The paragraph itself is necessary to explain the existence of the upward motion during summer seasons in the upper mesospheric altitudes.*

**Comment:**

**Confusion can also occur through mixing the use of vertical motions and vertical winds. For example, in the caption to Figure 1, the blue and red curves are labelled as "weighted mean vertical wind velocities" when they are actually vertical motions strongly weighted by the sedimentation speeds of the ice particles, i.e. not the vertical velocity of the neutral atmosphere. It is recommended that the terminology throughout the paper be corrected to ensure there is no misunderstanding of what is a vertical motion and what is a wind.**

Reply:

*We now use the term vertical velocities throughout the manuscript or describe why it can be interpreted as vertical wind.*

**Specific Comments**

**Comment:**

**P1, L12 "Disappearance" not "disappearing"?**

Reply: *"disappearance"*

**Comment:**

**P3, L15 Stoke's drift.**

Reply: *Accepted*

**Comment:**

**L4, L21 Brackets required around Hoppe and Fritts, 1995b**

Reply: *Accepted*

**Comment:**

**P5, L27 I do not understand "vertical shear amplitude of 5 m/s". Should this not be 5 s-1?**

Reply: *"Shear amplitude" here indicates a velocity difference between the given bins of the grid. We keep the terminology used in the referenced paper by Stober et al (2018a) and add the remarks in the text of the manuscript.*

**Comment:**

**P8, L22 I do not understand the sentence starting "They found ... " Is "continued' meant, rather than "preserved"?**

Reply: *"continued". „persisted" is used in the referenced paper.*

**Comment:**

**P9, L5 "threshold" rather than "point"?**

Reply: *"value". E.g. red circle above Δw=0.4 represents the average value of the velocity for all data points which uncertainty is lower than 0.4 (given value).*

**Comment:**

**P9, 14 "is" before "available".**

Reply: *Accepted.*

**Comment:**

**P9, L25 "reduced" rather than "slow down".**

Reply: *Accepted.*

**Comment:**

**P9, L26 "sediment due to gravity"**

Reply: *Accepted.*

**Comment:**

**P9, L32 "Presenting", not "pretending".**

Reply: *Accepted.*

**Comment:**

**P10, L5 Remove brackets around Berger and Lübken, 2015.**

Reply: *Accepted.*

**Comment:**

**P10, L34 "downward motions" not "the downwelling"?**

Reply: *Accepted.*

**Comment:**

**Figs. 2, 3 "Positive wind values correspond to"**

Reply: *Definitely.*

**Comment:**

**Fig. 4 Do the 2D histograms represent "vertical wind measurements" or 'vertical speed measurements?"**

Reply: *"Vertical velocity measurements". Please see the last reply in the general comments above.*

**Comment:**

**What do the dashed blue lines in Fig 4 represent?**

Reply: *Blue lines on Fig.4 are direct averages. It is now defined in the description of the figure.*

---

## Author Comment (AC2) · 24 Jan 2019

**Authors response to the reviewer#2 on "Are mean vertical velocities from PMSE a good representation of mean vertical winds?" by Nikoloz Gudadze et al.**

We thank the referee for her/his positive and helpful feedbacks. We appreciate all the efforts improving the manuscript and acknowledge the given comments and suggestions. We provide detailed replies to all the raised comments listed below. All changes in the manuscript will be highlighted as bold text.
The remarked language changes are also included.

**General Comment**

**Comment:**
**The description of the measurements is insufficient for readers not familiar with the topic. More details, and even basics of the MAARSY radar; e.g., the geographic coordinates are only provided in the abstract. More information is necessary here. Few more information on the experiments would also be helpful. Have the experiments been run continuously during both seasons? Which is the grid size for binning (page 2, line 5)? This and more information would improve the paper.**
> Reply:
> *We have extended information and description on the instrument and observational details.*

**Specific issues:**

**Comment:**
**P2 l 17: "...considering the Bousinesq approximation..." delete this part of the sentence. Continuity equation does not require incompressible fluid.**
> Reply: *Done*

**Comment:**
**P3, ll 22-24: The mechanism is described later, but should be outlined here.**
> Reply: *Done.*

**Comment:**
**P3, l 28: This sentence requires information that is only provided below or missing, namely the latitude of the observations, and an explanation what "PMSE 5 beam radial velocity" means.**
> Reply: *Information is added.*

**Comment:**

**P4, l 2: What does MAARSY stands for? It should not only be provided in the abstract.**

Reply: *Explanation of the abbreviation added.*

**Comment:**

**P4, l 22: more homogeneous than what?**

Reply: *Than within instantaneous measurements. Sentence is rephrased.*

**Comment:**

**P7 l 17: Jacobi, 2011 shows midlatitude winds, not polar ones. If you want to add more references, you may wish to refer to radar based wind climatologies like Portnyagin et al., 2004.**

Reply: *Thank you for the suggestion of interesting reference. We agree and changed accordingly.*

**Comment:**

**P8, l 16: ".. but not in certain circumstances associated with a target parameter." I do not understand what this means.**

Reply: *Reworded as* ".. but not in certain circumstances associated with a background conditions."

**Comment:**

**P9 l5 /Figures 5 and 6: I did not understand what the red points mean.**

Reply: *Reworded. Red points indicates mean of those vertical velocities corresponding to uncertainties lower than a given threshold.*

**Comment:**

**Figure 1: The blow-up on the right panels is not necessary in my opinion. The vertical resolution is 500 m and the effect is visible on the left panels also.**

Reply: *The main reason to zoom out the middle part of the right panels is to highlight the error bars given on red curve of the weighted averages. We add a sentences in figure description to point attention of the reader.*

**Minor comments**

**Comment:**

**P2, l 29: admitted -> considered**

Reply: *Accepted*

**Comment:**

**P6, l 7: excluding the -> except for a**

Reply: *Accepted*

**Comment:**

**P6 l 25: "We have also … " something went wrong with this sentence**

Reply: *Reworded*

**Comment:**

**P8, l 22: downward the -> the downward**

Reply: *Done*

**Comment:**

**P9, l 4: remove "relatively"**

Reply: *Done*

**Comment:**

**P 16, l 28: Stober et al., 2018: refer to the final revised paper**

Reply: *Done*

---

## Author Response (AR2)

Authors response on "Can VHF radars at polar latitudes measure mean vertical winds in the presence of PMSE?" by Nikoloz Gudadze et al.'

Authors response to co-editor

Dear Prof. Ward,

We are grateful to your decision on our manuscript.
We considered your comment to highlight the thermal wind balance for the zonal wind structure in the summer mesosphere. Corresponding text is added in the introduction. The change is given as a bold text on page 2, lines 13-15. Please find the enclosed last revised version of the manuscript.

Best regards,
N. Gudadze,
G. Stober,
J. L. Chau